# Model of Determining the Optimal, Green Transport Route among Alternatives: Data Envelopment Analysis Settings

**Luka Vukić [1,*], Tanja Poletan Jugović [2], Giambattista Guidi [3] and Renato Oblak [4]**

[1] Department for Maritime Management Technologies, Faculty of Maritime Studies, University of Split, Ruđera Boškovića 37, 21000 Split, Croatia

[2] Department of Maritime and Transportation Technology, Faculty of Maritime Studies, University of Rijeka, Studentska 2, 51000 Rijeka, Croatia; poletan@pfri.hr

[3] Department of Energy Technologies and Renewable Energy Sources, Italian National Agency for New Technologies, Energy and Sustainable Economic Development, Via Anguillarese 301, 00123 Rome, Italy; giambattista.guidi@enea.it

[4] Adria Polymers d.o.o., Poje 1, 51513 Omišalj, Croatia; renato.oblak@ri.htnet.hr

[*] Correspondence: lvukic@pfst.hr; Tel.: +385-98-549-849

**Abstract:** Data envelopment analysis (DEA) is a useful method for determining relative efficiency in many types of businesses, including the transport sector. In line with the European Union's (EU) policy of sustainable development of transport, external costs become the competitiveness factor of the transport route valorization. Presenting specific DEA settings, the paper aims to show and test a developed model for determining the optimal transport route among alternatives towards the same destination where external cost as a socio-ecological factor is included in DEA, along with transport cost (quantitative factor) and transport time (qualitative factor). In order to adhere to the principles of the least possible energy consumption, the given distance that also included in DEA settings represents the shortest route between the starting point and destination, as a unique and constant output variable. Therefore, the optimal direction selected by the DEA stands for the green route. The capabilities of the DEA, set up in this way within the broader model, are demonstrated in the practical case.

**Keywords:** DEA; decision-maker; green logistics; sustainable transport; transport route

---

## 1. Introduction

Environmental degradation, and in particular climate change, which is partly the consequence of activities in the transport sector, require rapid adjustment of European Union (EU) legislation supporting a trend of raising sustainability criteria, speeding up reforms, and encouraging, in particular, new EU Member States to reach the prescribed European standards [1]. These are the reasons why, in 2011, the European Commission adopted a new White Paper for the transport sector by 2020, which also defines measures for the long term, requiring a 60% reduction of greenhouse gas emissions in the transport sector by 2050 compared to 1990 [2]. The EU indicates that implementing the adopted standards is the obligation of utmost importance, and providing the concept of internalization of external cost is one of the most important transport policy tools to achieve the established objectives. Sustainable development policy guidelines have been transposed into national development strategies of the transport sector.

External costs of transport are costs arisen from its negative impacts on nature and society, such as traffic congestion, traffic accidents, noise, air pollution, climate change, technological processes of

production, distribution and consumption of oil and petroleum products and other energy (upstream and downstream processes), and infrastructure costs (infrastructure, wear and tear or pavement cost). The internalization of external costs represents monetary valuation and payment of damages.

Considering the aforementioned trends in the transport sector, regulations, and guidelines of the European Commission, it is evident that external costs of transport become an important factor in traffic planning and decision making, starting with the price, form, and direction of transportation, through segments in the chain of transport logistics services to strategic decision-making on the transport corridors. In this way, external costs imply an increasing influence on the intensity, dynamics, and direction of freight flows.

Such trends indicate the necessity of internalizing external costs considering them as decision-making factors in all business segments of the transport sector. The role of external costs in valuing and optimizing traffic routes as a decision-making segment within the transport network can no longer be ignored. The term "transport cost", considered conventionally as the cost of fuel, traffic vehicle, human work, etc., is now expanded, and should also include the external cost of transport.

External costs are not paid by either entity of the transport services chain, as the manufacturer of goods, seller, carrier, or buyer, and their size reaches 20% of the road transport cost and 11% of costs generated from railway transport [3]. The principles that "user pays" and "polluter pays" imply that market-based instruments involve these costs in the decision-making process of carriers and other stakeholders throughout the transport chain [4]. The route and form of transport service in the future will be decided to take into account the magnitude of external costs, based on market principles.

To date, external costs in the transport sector have been given much consideration, but there is a clear need to analyze these costs for each mode of transport and each traffic route. In the context of external costs that significantly affect society and quality of life, it is important to objectify the need to use an alternative, green mode of transport. External costs are an important indicator of a country's transport and economic development, pointing to guidelines and trends in line with high-level strategic documents [5]. Consequently, it can be envisaged that without calculating the external costs of the transport route, its competitiveness in the transport services market could not be completely assessed.

Environmental pollution, traffic congestion, and accidents are the main sources of external costs in traffic. By analyzing the external cost structure of maritime and rail transport, it is possible to identify the lack of two of the three main structural elements mentioned, the external cost of congestion and traffic accidents, which together account for 60–70% of overall external costs [6,7]. This means that the third major structural element, environmental external costs, and the other external costs of marginal significance (in terms of total share), are the main indicators of the adverse impact of maritime and rail transport.

Emissions directly (about 1/3 of the share) and indirectly through climate change impacts (about 1/3 of the share), up and down streaming processes (about 9% of the share), negative impacts on biodiversity and agrarian cultures, and urban areas are responsible for more than 80% of external costs in maritime and rail transport. Other external costs, such as noise and infrastructure costs, do not together exceed 10% of the total external costs [6,7]. Considering that emissions are a product of the combustion of fuel, and that the magnitude of emissions and fuel consumption are easily and objectively measured, this knowledge can be crucial in the decision-making process of the logistics operator in the choice of the most convenient form and route of transport. According to the socio-ecological criterion, green transport shows the least harmful impact on the environment and health, thus, the freight, intermodal, maritime-rail transport (electrically operated railway) is the optimal form, and the route of the first choice is the shortest one. That choice is a result of the usual absence of congestion and accidents in these kinds of transport, as well as the low environmental external costs of electrically operated railway, and the maritime routes passing beyond populated areas.

The introduction of ecological criteria in the process of valorization of traffic routes, also including a transport mode, has been the subject of scientific research, as well as scientific projects, for the last ten years. Following the existing literature, it can be seen that external costs in the transport sector are

becoming increasingly important, and a factor that cannot be ignored. The aim of the project MARCO POLO [8] was to reduce road congestion and environmental pollution by encouraging the relocation of road transport to 'greener' modes of transport, such as rail, sea, and inland waterway transport. The project advocates the principles of sustainability: each process must have economic, social, and environmental significance and cost-effectiveness. In [4], the authors advocate the internalization of external costs as one of the key tools for the implementation of EU policy in the transport sector, emphasizing that external cost is an important factor in the decision-making of all stakeholders in the logistics supply chain. In [9], the authors examined possible changes in the mode of transport on traffic routes from Greece to Northern German destinations under the condition of declaring the Mediterranean Sea as a Sulfur Emission Control Area (SECA). They used a model with two variables, transport cost and time, while the environment variable of the external cost was calculated separately with EcoTransIT software. The results show a modal shift towards road transport, and this unexpected fact confirms the need to valorize each transport route. The paper encourages thinking about incorporating the external cost variable within multicriteria analysis as a continuous influential variable. In [10], the author examines the external costs of intermodal transport on the transport route, and proves that it does not have to be environmentally friendly compared to other forms of transport if the length of the transport route, the size of the maritime part of intermodal transport, and speed in road transport are not taken into account. Knowledge of alternative and environmentally friendly transport routes by logistics operators is required. The need to reduce external costs indirectly indicates the need to optimize cargo flow. In a study prepared for the U.S. Congress [3], a significant share of external costs in total freight costs was pointed out. Within the current trends of green logistics [11], corporate responsibility in port operations is being developed to incorporate environmental factors into strategic development plans. Written norms and unwritten rules strengthen the competitiveness of ports based on environmental factors as competitive factors of the business. In [12], the authors point out external costs as a current, key topic in the transport sector, and the European Union's commitment to affirm external costs as a competitive factor in freight transport to implement the set policy. They list external cost research methods, as well as mathematical models. In [13], the necessity of reducing external costs through a new logistics supply chain design is emphasized, which, in addition to new infrastructure, envisages alternative modalities of transport, and also new locations of ports and hubs in the hinterland, which indirectly indicates the impact of external costs on optimizing traffic routes and flows. Sustainability in the economic, environmental, and social dimension, as well as continuous improvement of maritime safety, are the main goals of the project HORIZON 2020 [14], and the project CEF [15] is implemented to connect people and create new jobs in the sectors of sustainable transport, renewable energy sources, reducing $CO_2$ emissions and digital technology. In the transport sector, it supports multimodal transport, intelligent transport systems, new technologies, new corridors, and freight flow within the framework of sustainable development. Using multi-criteria analysis [16], the authors ranked the models of the system of motorways of the sea in Croatian ports taking environmental criteria, including external costs as decisive, and in line with transport development policy trends in the European Union. Transport sustainability issues currently occupy the scientists dealing with topics such as a modal shift towards maritime transport [17], the selection of green intermodal chains [18], or the development of cost models for urban transport infrastructure options [19], all to reduce social and external costs.

Based on all the presented facts in this paper, the authors aim to develop a model of transport route valorization, which would use a well-known multi-criteria method that is easy to handle, that uses basic elements of valorization including environmental criterion, and is easily extensible. According to the set criteria, the authors opted for a data envelopment analysis (DEA) whose settings they specifically designed for this purpose. The model considered in the presented case study assumes two basic facts: internalization of all external costs and approximately equal quality of transport service available on all examined traffic routes.

## 2. Materials and Methods

This paper examines the competitiveness of intermodal, maritime-rail transport of a TEU (twenty-foot equivalent unit) weighing 10 t on transport routes, having a port of origin in Shenzhen (CHN) via the Suez Canal to selected Central European destinations, as follows: Budapest, Vienna, Prague, and Munich. Selected routes include eight different intermodal hubs (ports): Trieste, Koper, and Rijeka in the Northern Adriatic area, Genoa and Thessaloniki in the Mediterranean, Constanta in the Black Sea, and the northern European ports of Rotterdam and Hamburg. This way, eight traffic routes were formed for each Central European destination, with the main objective of determining optimal traffic routes in the context of the simultaneous influence of quantitative (transport costs), socio-ecological (external costs), and qualitative (transport time) criteria. The research was conducted in 2018, and all costs were calculated based on prices that were valid that year.

Comparative analysis of the transport routes is based on specific parameters, namely: transport costs per unit (TC), external costs per unit (EC), transport time (t), and distance (s). The data required for the DEA has been prepared with the appropriate software and the technical sheets.

Emissions, up and down streaming processes, and climate change impacts of intermodal, maritime-rail transport on selected routes are calculated using the EcoTransIT® World software (Update 30th June 2016) [20], as well as the transport time (t) and length of the transport route (s). In choosing the type of transport and unit values of pollutants, the authors were mainly guided by the principle of the lowest possible external costs, minimizing the subjectivity in the selection, and emphasizing that the real values in practice are always higher. GHG (greenhouse gases) emission expressed as an equivalent emission of $CO_2$ ($CO_{2eq}$), as well as SOx, NOx, PM (particulate matters), and NMVOC (non-methane volatile organic compounds) emissions were included. The research plan adjusted to a lesser extent to the limitations of the computer program in the parts related to the set of input data necessary for the operation of the program, as well as redefining sections of traffic routes. In the calculation of external costs, the location of loading at the departure point and transshipment at the intermodal hub is very important, because the correct location reduces the share of road or rail diesel transport to the loading or transshipment terminal. Namely, the computer program has its automated procedures that determine the exact location for loading/reloading of goods at the port or railway terminal. These mechanisms cannot be influenced except by changing the loading/transshipment location. Thus, for example, the lesser-known port of Shekou in China, which is one of the ports of Shenzhen, was chosen as the starting point. This procedure reduced the share of road freight transport in Shenzhen to a minimum (8.89 km), compared to the length of road transport normally determined by the computer program for this city. By this procedure, more realistic (lower) values of external costs can be obtained in the research model, because the previously mentioned forms of transport can significantly increase them on shorter sections as well. The location of the destination is chosen by the program itself (usually the railway station), equally for each destination, which is a sufficient guarantee of comparability of results. In the research model, the transport ends at the place of unloading.

External costs in maritime transport refer to the type of ship CC Intra-Continental non-EU capacity 0.5–3.5 kTEU. The selection was made based on software solutions that this type of ship is burdened with the lowest external costs, and an emission calculation for that ship in each selected port in the research exists.

External costs in rail transport, by an electric railway, were calculated for a container train with a capacity of 1000 t, LF (load factor) 49.8%, and ETF 20% (empty trip factor), according to the suggestion of a computer program.

The EcoTransIT software has been tested for a year and found that it is simple, reliable, and elaborated in detail but not always up-to-date. The results of emissions that depend on the amount of freight led through the model to the same final results, so, for better transparency in the research, the unit freight is manipulated.

The handbook on external costs of transport by [7] has been used in the monetary valuation of obtained data [Table 1]. Published emission values for maritime transport are limited, and are lowest

in the Mediterranean Sea. In order to simplify the calculation process and manipulate the lowest emission values, they were also used for calculation in the Black Sea, the North Sea, and the Atlantic, where emission values are slightly higher and in the Red Sea, the Indian Ocean, and the South China Sea, where they are slightly lower.

**Table 1.** Unit prices of pollutants used in the research [7].

| Air Emission Rail Transport | 2010 €/kg | Reevaluation 2018 + 10.22% (2011–2017) |
|---|---|---|
| $CO_2$ | 0.09 | 0.0992 |
| SOx | 10.24 | 11.29 |
| NOx rural | 10.64 | 11.73 |
| $PM_{2.5}$ rural | 28.11 | 30.98 |
| $PM_{10}$ average | 4.50 | 4.96 |
| NMVOC | 1.57 | 1.73 |
| **Air Emission Maritime Transport** | **2010 €/kg** | **Reevaluation 2018 + 10.22% (2011–2017)** |
| CO2 | 0.09 | 0.0992 |
| SOx | 6.70 | 7.38 |
| NOx | 1.85 | 2.04 |
| $PM_{2.5}$ sailing mode | 18.50 | 20.39 |
| $PM_{10}$ average | 2.96 | 3.26 |
| NMVOC | 0.75 | 0.83 |

Unit emission values for the period 2011–2017 (7 years) were revalorized in line with EU GDP growth according to Eurostat with a rate of 1.46% per year [21].

External costs values of the intermodal hub ports (external costs of the ship in port and transshipment of unit cargo) have been taken from the literature as constants. [22].

The cost of maritime transport for selected routes is calculated using the World Freight Rates software [23], and the cost of rail transport by the Sea Rates software [24]. The cost of the intermodal hub port is calculated based on official terminal tariffs and data provided by the agents at the target ports (costs marked as other). The research flow chart is shown in Figure 1.

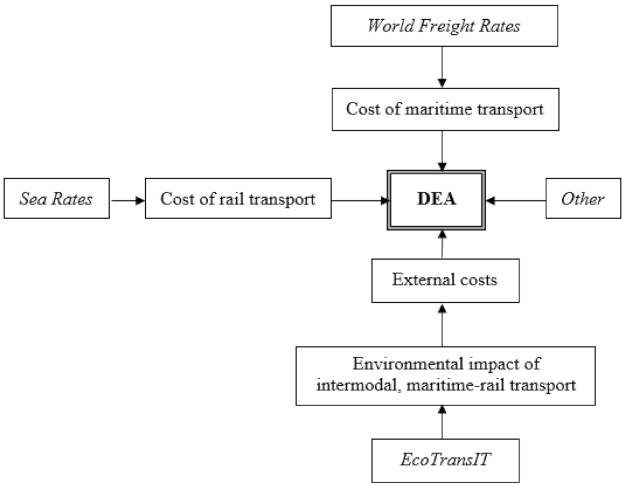

**Figure 1.** Flow chart of the research.

Data are processed using the data envelopment analysis (DEA) supported by Frontier Analyst Banxia Software, version 4.3.0. [25]. DEA uses transport costs (TC), external costs (EC), and transport time (t) as input values, confronting them with the given distance of transport route ($s_g$) as the output

value. The given distances represent the shortest traffic routes from Shenzhen to the four Central European destinations. The data set (2018) used in the DEA is shown in Table 2.

**Table 2.** Transport costs per unit (TC), external costs per unit (EC), transport time (t) and distance (s) on the maritime-rail route from Shenzhen (CHN) to Central European destinations.

| Budapest via | TC (€) | EC (€) | t (Days) | s (km) | Prague via | TC (€) | EC (€) | t (Days) | s (km) |
|---|---|---|---|---|---|---|---|---|---|
| Constanța | 3084.28 | 320.56 | 26 | 15,109 | Constanța | 4004.08 | 372.25 | 27 | 15,695 |
| Thessaloniki | 3107.21 | 359.19 | 25 | 14562 * | Thessaloniki | 3888.64 | 377.60 | 26 | 15,142 * |
| Rijeka | 2642.08 | 344.16 | 26 | 14,989 | Rijeka | 3249.10 | 348.65 | 27 | 15,255 |
| Koper | 2754.06 | 348.11 | 26 | 15,153 | Koper | 3207.45 | 350.98 | 26 | 15,349 |
| Trieste | 2749.18 | 347.98 | 26 | 15,141 | Trieste | 3185.47 | 350.96 | 26 | 15,353 |
| Genoa | 3246.41 | 367.95 | 27 | 15,859 | Genoa | 2773.48 | 368.86 | 27 | 15,878 |
| Rotterdam | 4031.03 | 456.67 | 33 | 19,560 | Rotterdam | 2959.02 | 452.08 | 32 | 19,118 |
| Hamburg | 3451.03 | 467.70 | 32 | 19,835 | Hamburg | 2700.84 | 450.81 | 31 | 19,281 |
| **Vienna via** | **TC (€)** | **EC (€)** | **t (Days)** | **s (km)** | **Munich via** | **TC (€)** | **EC (€)** | **t (Days)** | **s (km)** |
| Constanța | 3474.69 | 359.01 | 26 | 15,363 | Constanța | 4166.87 | 366.83 | 27 | 15,821 |
| Thessaloniki | 3440.95 | 364.43 | 25 | 14,809 * | Thessaloniki | 4482.84 | 368.68 | 26 | 15,194 |
| Rijeka | 2790.01 | 340.15 | 26 | 14,997 | Rijeka | 2701.29 | 341.87 | 26 | 14,989 * |
| Koper | 2764.65 | 342.58 | 26 | 15,090 | Koper | 2719.06 | 344.32 | 26 | 15,082 |
| Trieste | 2754.88 | 341.55 | 26 | 15,070 | Trieste | 2677.55 | 344.27 | 26 | 15,033 |
| Genoa | 3168.26 | 371.54 | 27 | 15,702 | Genoa | 2244.39 | 357.41 | 26 | 15,426 |
| Rotterdam | 3656.60 | 451.15 | 33 | 19,310 | Rotterdam | 2853.20 | 446.40 | 32 | 18,972 |
| Hamburg | 3046.48 | 463.10 | 32 | 19,640 | Hamburg | 3016.36 | 453.07 | 32 | 19,404 |

* given distance. TC = sum of the costs of maritime transport, rail transport, and intermodal hub (unit cargo transshipment costs in the port from ship to the terminal and from terminal to rail), EC = sum of external costs of maritime transport, rail transport, and intermodal hub (external costs of the ship in port and transshipment of unit cargo), t = sum of marine and rail transit times, s = sum of lengths of sea and rail routes.

## 3. DEA Settings

Data Envelopment Analysis (DEA) is a deterministic, non-parametric method that determines the relative efficiency of a decision-maker (DM) in a manufacturing or non-manufacturing sector [26]. A decision-maker is a responsible person who has the right and duty to manage the business process. Non-parametric features of the method, which include samples of different quality and strength, and often do not show normal distribution patterns and variance, variables from daily business processes not qualitatively suitable for statistical analysis, or multiple variables that are not mutually comparable [27], have their statistical weaknesses, but firmly determine business efficiency. This is why DEA is a sovereign, affirmed and one of the most commonly used economic multi-criteria methods that are applied when the possibilities of statistical methods are depleted. It has proven successful in evaluating the supply chain [28] and environment-related efficiency [29]. Efficiency is a principle that is measured and evaluated in the business process, and represents the success of converting invested resources and work into business results. Efficiency increases when business results are higher, and investment is lower. In the transport sector, in the narrower sense, efficiency is higher when, e.g., costs are lower and transport times are shorter by the same distance traveled, or the case when a longer path is crossed for the same cost and at the same time. The principle of efficiency is expressed by comparing operating results (weighted sum of outputs) and investments (weighted sum of inputs), in order to achieve maximal business success. The most commonly used variant of the DEA, the CCR (Charnes, Cooper, and Rhodes) model, is usually displayed as a fraction [30,31]:

$$Max\ h_k = \frac{\text{outputs}}{\text{inputs}} = \frac{\sum_{r=1}^{s} u_r y_r}{\sum_{i=1}^{m} v_i x_i} \tag{1}$$

under condition that

$$\sum_{r=1}^{s} u_r y_r \leq \sum_{i=1}^{m} v_i x_i \tag{2}$$

respectively

$$\frac{\sum_{r=1}^{s} u_r y_r}{\sum_{i=1}^{m} v_i x_i} \leq 1 \tag{3}$$

and

$$u_r \geq 0, \quad r = 1, 2, \ldots, s; \, as \, well \, as \, \, u_r \geq \varepsilon \tag{4}$$

$$v_i \geq 0, \quad i = 1, 2, \ldots, m; \, as \, well \, as \, \, v_i \geq \varepsilon \tag{5}$$

where $h_k$ = relative efficiency of $k$ (DM); $k$ = number of DM units; $y_r$ = number of output $r$; $u_r$ = weight allocated to output $r$; $x_i$ = number of input $i$; $v_i$ = weight allocated to input $i$, $\varepsilon$ = small positive value (mostly $10^{-6}$).

The result follows as

$$0 < h_k \leq 1 \tag{6}$$

where the result $h_k = 1$ means that the $k$ (DM) is relatively efficient, and the results $h_k < 1$ mean relative inefficiency of the $k$ (DM), showing the value how much the output/input ratio has to be improved to reach maximal relative efficiency of that $k$ (DM). Relative efficiency represents the level of realization of the virtual, maximum efficiency determined by the DEA method.

DEA is a method by which one can determine the best DM between different DMs on the same or similar business task with multiple inputs and output parameters. Following the principles and capabilities of the DEA method, this study examines the relative efficiency of DMs on a traffic route to a particular destination, determining specifically which of the analyzed traffic routes to the preferred destination shows the highest efficiency and, by this criterion, represents the route of choice. In the variant of combining minimum inputs to reach the same output (input-oriented model of DEA), the input values in this research mandatory include transport costs, external costs, and transport time. Given that this is research in the transport sector, the output value is the traveled distance as a targeted service of the business sector.

The features of the DEA concerning the specificities arising from the characteristic of the transport industry need to be clarified. Although the relation between input and output variables is generally not important for DEA, it is necessary to satisfy the condition of isotonicity, i.e., that a positive change in input results in a positive change in output, or staying the same at least [32]. This condition, applied to the conception of this research, represents a contradiction, interpreting that an increase in transport costs, external costs, and transport time (present input variables) increases the transport route (output variable). In this concept, dependent and independent variables were substituted, an operation which is not possible in statistical analysis, but in non-parametric analyses such as DEA is. There are at least two explanations and justifications for this action: an input-oriented model of DEA and a single and unchangeable (constant) variable of output. Transport costs, external costs, and time represent an investment in the transport of goods to the desired destination. This input-oriented model endeavors to achieve the aim with as little investment as possible. In the transport industry, the transport route is a product; in commercial activity is a resource. The transport route is, therefore, the output in the transport sector; while, in some other industries, it can be defined as the input, a cost. As an output component, the distance in the transport industry can also be expressed as energy consumption [33], which is the result and consequence of transport. It harms efficiency and seeks to minimize it. In this model, the transport route is a fixed variable. Dependence relation between input and output components remains the same without the possibility to change. The transportation route can also be shown as a function of the sales value of the goods in the output and by including the production value of the goods in the input. In that constellation, a longer route with higher costs and longer travel time results in lower revenue. By testing DEA, it can be defined that, under the aforementioned conditions, the efficiency results on the proposed traffic routes are the same, and that differences in efficiencies depend on the length of the transport route. Therefore, the results can be obtained by the simple model that is proposed.

There are no firm default frameworks for how to select input and output variables, and they are usually formed in collaboration with economic and mathematical experts, according to the research objective [34]. In line with the aim of this research, the given distance is a key value for DEA and represents the least energy consumption variable. The basic principle of the least possible energy consumption on a transport route to a particular destination meets the requirement of the least possible impact of transport on the environment and human health. Thus, all output values are the same in the form of the shortest distance so, instead of the real value, a positive constant (> 0) can be theoretically used, e.g., 1. Since it is a constant and unique output parameter, it cannot affect the isotonicity rule. A single variable output cannot be affected by another one as it does not exist; the ratio between changeable input variables and constant output value depends on the input only. The distance traveled as a task to be performed by the transport activities on the traffic route, is included in the DEA as the output value. In the method for examining the efficiency, the optimal distance (virtual output) is the shortest possible distance between the starting point and destination via targeted intermodal nodes (ports) in a given intermodal chain, appointed in this research as a given distance. In the context of external costs, the shortest possible distance is the given value of the least energy consumption required to complete the task. Therefore, the given distance is a fixed value of the output as an ultimate goal achievable with different values and variants of inputs to determine the optimal traffic routes to a particular destination where the input (transport costs, external costs, and transport time) is minimal. Finally, in this research, the DEA settings are shown by the formula:

$$\frac{s_g}{TC, \ EC, \ t} \tag{7}$$

where the given distance ($s_g$) as the output is faced with transport cost (TC), external cost (EC), and transport time (t) as inputs.

Although the method requires the normalized input and output values (the quotient of tested and highest values), in this research, it shows the same results with the real values, and in that form are presented.

Discussion on controlled (discretionary) and uncontrolled (non-discretionary) input values is necessary to explain the choice among the options offered by the software. There is no transport without transport costs, external costs, or transport time; in this sense, one cannot be influenced by whether or not these items exist. However, the decision-maker can discretionarily influence the magnitudes of the input values. The choice of carrier, negotiation skills, information, and reputation in the market of transport services will influence the final amount of transport costs, so this input value is controlled by the DM actions. Although seemingly the size of the external costs cannot be influenced (in transportation, not in the technological sense), as they occur as a result of the performance of transport service itself, indirectly, by choosing a shorter route and appropriate modality of transportation, the decision-maker reduces the size of the external costs also controlling this input value. The time of transport directly depends on the type and route of transport, and the decision-maker in the transport sector can influence it by appropriate choice among alternatives. However, the decision-maker can, more or less, control the magnitudes of the transport costs and length of the transport route, but has no discretionary possibility to predict and manage events on the route during transport. Transport time is part of a business or contractual relationship, but a guarantee of realization does not depend entirely on the intention of the service provider so, it is marked as uncontrolled value. The consequence of this decision is that transport time as an uncontrolled input variable within DEA has a lower impact on the ultimate efficiency of the examined traffic route than other controlled input variables. Given the correlation between external costs and transport time, this fact reduces the significance of the external cost variable, which is in line with the principle to minimize external costs at all stages of the research, to reduce their possible preferential position in it.

The final result of the DEA processing is the ranking of different traffic routes towards the same destinations by the relative efficiency criterion determined by the combined effect of transport costs,

external costs, and transport time. This study presents the results of the CCR model of the DEA, in which the input and output values behave linearly. Efficiency in the CCR model represents both the scale and technical efficiency [26]. A graphical presentation of the input-oriented form of the DEA in the CCR variant is shown in Figure 2.

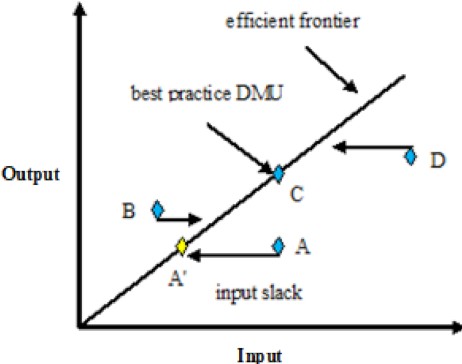

**Figure 2.** The concept of the Charnes, Cooper, and Rhodes (CCR) input-oriented model of data envelopment analysis (DEA) (single-input and single-output variant).

In the schematic view of the simplified variant, the DEA efficiency frontier is a line and the point at the line represents the decision-making unit (DMU) of maximal efficiency. All other DMUs moved away from the line are inefficient. In the input-oriented model, the DMU inputs should be corrected to become efficient, and their horizontal distance off the line represents the size of correction required (input slack). In the two-input and single-output variant, the efficiency frontier is a curve, and inefficient DMUs require correction, which is measured as the DMU point distance off the curve ($\overline{AA\prime}$) at the line passing through the DMU point and the zero point (Figure 3). In this example, both inputs should be reduced.

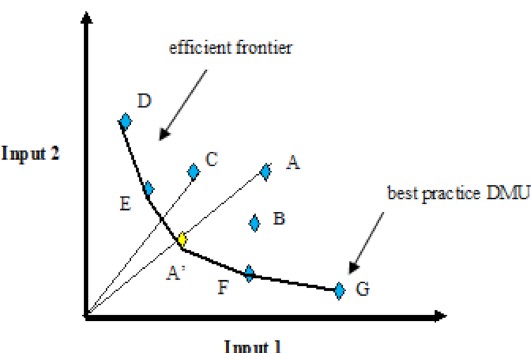

**Figure 3.** Efficiency frontier in the DEA model with two-input and single-output variant.

In the variants with multiple inputs and outputs, the variables are mutually analyzed and can be shown by a series of graphs.

A more detailed analysis of the DEA results reveals in which segment and to what extent the improvements in business efficiency are possible. At the same time, this is the guideline for DM indicating which segment of the business and to what extent it should be corrected. The coefficients of efficiency as general items, and particularly, the magnitude of possible improvements of the external cost items, show the impact of internalization of external costs on the efficiency of the transport business in this model.

## 4. Results

Results of the DEA are shown in the form of ranking the alternatives according to the criterion of relative efficiency (Table 3). DEA also shows a level of possible improvement for each variable of the alternatives ranked below the maximum efficiency (Table 4).

**Table 3.** Transport route rankings among the alternatives from Shenzhen to Central European destination, according to DEA.

| Budapest via | Score | Efficient | Vienna via | Score | Efficient |
|---|---|---|---|---|---|
| Constanta | 97.30% | FALSE | Constanta | 94.70% | FALSE |
| Genoa | 93.50% | FALSE | Genoa | 91.60% | FALSE |
| Hamburg | 74.80% | FALSE | Hamburg | 88.50% | FALSE |
| Koper | 98.90% | FALSE | Koper | 99.50% | FALSE |
| Rijeka | 100.00% | TRUE | Rijeka | 100.00% | TRUE |
| Rotterdam | 75.40% | FALSE | Rotterdam | 75.40% | FALSE |
| Thessaloniki | 100.00% | TRUE | Thessaloniki | 100.00% | TRUE |
| Trieste | 98.90% | FALSE | Trieste | 99.80% | FALSE |
| **Prague via** | **Score** | **Efficient** | **Munich via** | **Score** | **Efficient** |
| Constanta | 93.70% | FALSE | Constanta | 93.20% | FALSE |
| Genoa | 100.00% | TRUE | Genoa | 100.00% | TRUE |
| Hamburg | 99.90% | FALSE | Hamburg | 77.80% | FALSE |
| Koper | 100.00% | FALSE | Koper | 99.30% | FALSE |
| Rijeka | 100.00% | TRUE | Rijeka | 100.00% | TRUE |
| Rotterdam | 92.50% | FALSE | Rotterdam | 79.70% | FALSE |
| Thessaloniki | 92.90% | FALSE | Thessaloniki | 92.70% | FALSE |
| Trieste | 100.00% | TRUE | Trieste | 99.30% | FALSE |

**Table 4.** Potential opportunities to improve efficiency on transport routes from Shenzhen to central European destinations, according to DEA.

| Budapest via | TC % | EC % | t % | Vienna via | TC % | EC % | t % |
|---|---|---|---|---|---|---|---|
| Constanta | −13.3 | −2.7 | 0 | Constanta | −18.6 | −5.3 | 0 |
| Genoa | −19.1 | −6.5 | −3.7 | Genoa | −12.7 | −8.4 | −3.7 |
| Hamburg | −25.2 | −26.4 | −18.8 | Hamburg | −11.5 | −26.5 | −18.8 |
| Koper | −5.2 | −1.1 | 0 | Koper | −0.5 | −0.7 | 0 |
| Rotterdam | −35.1 | −24.6 | −21.2 | Rotterdam | −24.9 | −24.6 | −21.2 |
| Trieste | −5.1 | −1.1 | 0 | Trieste | −0.2 | −0.4 | 0 |
| **Prague via** | **TC %** | **EC %** | **t %** | **Munich via** | **TC %** | **EC %** | **t %** |
| Constanta | −18 | −6.3 | 0 | Constanta | −34 | −6.8 | −3.7 |
| Hamburg | −0.1 | −18.2 | −12.9 | Hamburg | −22.2 | −22.2 | −18.8 |
| Koper | −0.7 | 0 | 0 | Koper | -2 | −0.7 | 0 |
| Rotterdam | −7.5 | −18.4 | −15.6 | Rotterdam | −20.3 | −20.3 | −18.8 |
| Thessaloniki | −16.9 | −7.1 | 0 | Thessaloniki | −39 | −7.3 | 0 |
| | | | | Trieste | −0.7 | −0.7 | 0 |

Optimal routes towards central European destinations, selected within the proposed research model by DEA, are shown in Figure 4.

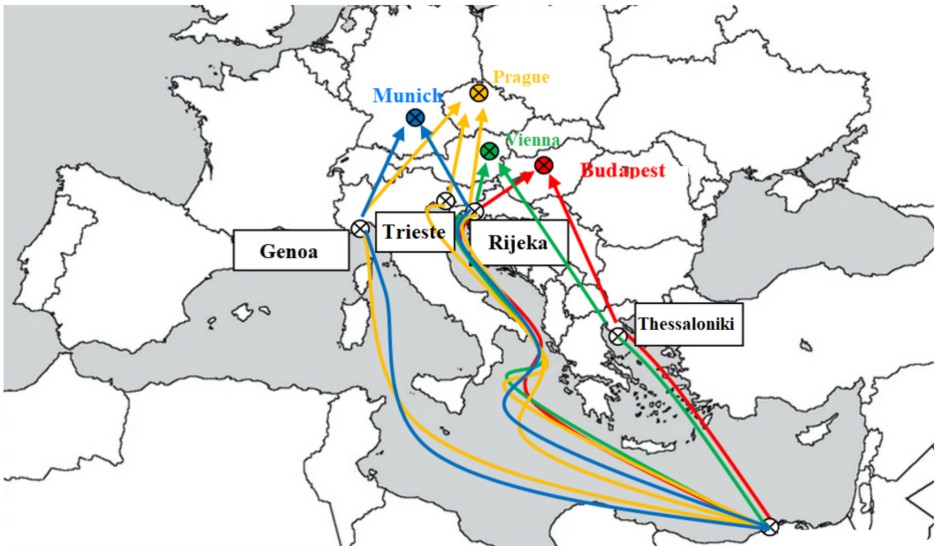

**Figure 4.** Selection of optimal traffic routes, according to the set research model [35].

The combination of variables is well-balanced for analysis, as it contains representatives of the quantitative, qualitative, and environmental factors within the input values. It also complies with the prescribed condition [36] that the number of DMs examined is several times greater than the sum of the variables examined (two times here), so these results can be considered a credible source of information.

Within the set model of research on traffic routes to Budapest, the DEA determined two optimal traffic routes, across Rijeka and Thessaloniki. Analyzing further the proposed combination, one can notice the index of 99% of the efficiency of other Northern Adriatic ports as well as the inefficiency of Northern European ports. Based on the set model, it is evident that all three input variables (TC, EC, t) on traffic routes via Rotterdam and Hamburg require improvements in high percentages to achieve optimum efficiency. The cost of transport is determined by the market (it is necessary to reduce the cost of transport by 35.1% on the route via Rotterdam and 25.2% via Hamburg), and the transport time can be reduced (without increasing transport cost) only by changing (shorting) the traffic route. This would optimally mean a reduction of external costs by 24.6% over the Rotterdam route and 26.4% over the Hamburg route.

Similar results are obtained on the traffic routes towards Vienna, where the model defines the traffic routes over Rijeka and Thessaloniki as optimal, and the other Northern Adriatic ports are more than 99% efficient. Traffic routes over Rotterdam and Hamburg are indicated as inefficient. Analyzing the possibilities for improving efficiency, it can be seen that the value of the cost of transport via Hamburg is lower than in all other routes to Vienna except via Northern Adriatic ports (unlike the route via Rotterdam which requires a 24.9% reduction in the parameter of transport costs). Examining the values of external cost items, the results are similar to those on routes towards Budapest. According to the model, the external cost values should be reduced by 24.6% and 26.5%, respectively, on the routes via Rotterdam and Hamburg to become efficient.

On the traffic routes towards Prague, the results of the defined model indicate traffic routes via Genoa, Trieste, and Rijeka as optimal. The traffic routes over Koper and Trieste are 99–100% efficient, as well as the route via Hamburg. The routes over Rotterdam, Thessaloniki, and Constanta have proved as inefficient. While the traffic routes via Thessaloniki and Constanta show a need to reduce transport costs by 16.9% and 18%, respectively, the routes via Hamburg and Rotterdam require an external cost reduction of 18.2% and 18.4%, respectively.

The results of the research are within the model determine the optimal traffic routes towards Munich over the hub ports of Genoa and Rijeka. Directions over the other Northern Adriatic ports are more than 99% efficient. The routes via Rotterdam and Hamburg show weak results. Analyzing the

possibilities of improving efficiency, one can identify the equal demands for reducing transport and external costs, for the route via Hamburg by 22.2% and via Rotterdam by 20.3%.

## 5. Discussion

Data obtained using this model assist the decision-maker to choose the optimal traffic route, considering the quantitative, qualitative, and environmental criteria. The model promotes the principle of the shortest possible route of transportation as the ultimate goal of all three criteria in the planning of freight flows and, thus, respects the principle of sustainability. Therefore, the optimal traffic route generated by the model is characterized as the "green" route. It can be used by any logistic chain stakeholder who is interested in simultaneously reducing transportation costs, external costs, and transportation time. The model examines the traffic routes for freight transport at the given distance in the shortest time, having the lowest transport and external costs, while simultaneously estimating and ranking the efficiency level of the observed traffic routes from 0 to 100%. It should be emphasized that this model was created as an economic, rather than a mathematical model, and that the shortest transport route may not necessarily be the most efficient one, justifying the purpose of multi-criteria analysis. The selection of the implemented parameters depends on the priority, discretion, and interest of transport chain stakeholders, so implementation of the diverse parameters instead of those used in this model is possible. In a very similar DEA model, the [37] set a value of 1 as a constant in the output. Although set up correctly, there is no elaboration on why the model works. Among the others, the distance variable in the input is not specially highlighted in that model, and there is a need to use another method to calculate the shortest distance. In our model, the given distance in the output, chosen by EcoTransIT, is prominent, visible, clear, and easily explained. It serves the motivation of the decision-maker to follow the principles of green logistics, and also as an argument to justify the chosen transport route in a case when such a decision has strategic importance and causes significant financial effects and consequences.

Comparing the DEA with the MCDA (multi-criteria decision analysis), the authors do not give a preference for any of them, and believe that there are not good and bad methods. The methods should be well-known and chosen correctly. In all of them, the settings are crucial. In the Promethee method, as one of the representative kinds of MCDA [16], detailed analysis of a series of criteria and sub-criteria performed, the weighting can be uncertain, and the procedure seems complicated and unsuitable for daily use. The DEA method looks like simpler and faster, especially this proposed variant manipulating only quantitative variables. With the MCDA method, problem analysis is probably more in-depth, but sometimes more subjective. DEA determines efficiency, while MCDA has a wide range of uses.

A limitation of the research is in missing the updated version of the handbook on external cost in transport, not used because it was published too late to fit into the data set. However, it shows a cumulative increase in unit prices of 8%, which cannot affect the conclusions except strengthening the importance of external costs even more. Due to by far the largest share of $CO_2$ in total emissions in transport (from approximately 40% in maritime transport to approximately 75% in electric rail transport), the growth of the newly published and revalued unit price of $CO_2$ of only 3.31% in the year 2018 (from 0.0992 EUR/kg, according to the old edition to 0.1026 EUR/kg, according to the new one) confirms that conclusion [38].

The research is also limited by the performances of EcoTransIT software: the more advanced the program, the more objective the results will be. The model does not need to use this software, but it allows for easier handling. Uncertainties are common on the transportation route. Failures, weather conditions, traffic congestion, unloading/reloading times, accidents, etc., happen and often are not predictable. Even if they are, they usually cannot be avoided. Uncertainties burden all traffic routes, but some of them can temporarily be burdened more, which the authors or logistic providers may not always know. The transport time variable is uncertain, and sometimes does not correlate with the distance variable. EcoTransIT cannot count on uncertainties. For reasons of uncertainties, the assumption was introduced in the conditions of the research that the quality of transport service is

approximately equal on all transport routes. In the DEA settings, the transport time variable is marked as an uncontrolled one and, thus, less burdensome on the other variables encountered in the method.

Adaptation to new living and working conditions, necessary to maintain the standard of modern life, is possible only by respecting the principles of sustainability. Green logistics in the transport sector promotes adherence to these principles at all levels of the transport chain network considering environmental elements in every individual decision-making procedure. The slowdown of global warming by decreasing the role of the transport sector in the segment of the environment pollution, has become a principle without which transport policy cannot be imagined today [39]. Green legislation is being introduced worldwide and even into the humanitarian supply chain, following the principle of the least possible environmental damage. It is crucial to reduce the dependency on the oil of the transport system without detriment to its efficiency.

In the long-term, the transport sector policy will promote green, safe and clean transport. With stimulating and repressive measures, this policy will encourage changes in the directions of goods flows, directing them towards green routes, while moving them away from traffic congestions and emissions harmful to human health. External costs are becoming a competitive factor and many traffic routes, where freight flows are currently inefficient and of low intensity will get the opportunity to increase competitiveness. Qualified, trained, and motivated logistics operators as decision-makers in the logistics supply chain should get a central role and responsibility in this strategic shift. The policy of sustainable transport development is also to be implemented by other stakeholders in the logistics transport chain, including shipping companies and ports. A prerequisite for this is the complete internalization of external costs, which was one of the basic assumptions of this research. In such circumstances, all stakeholders have the interest to create opportunities to participate in the green corridor.

The European Union's transport policy and its increasingly demanding objectives indicate that the so-called socio-ecological criterion, representing all the negative impacts of transport on nature and society, being monetized and presented as external costs, is becoming an increasingly strong indicator of the valorization of freight flows on transport routes.

In the provision of transport services, the quantitative (economic) and qualitative criteria currently prevail in the decision-making process, while external costs have not been decisive so far. This new criterion will change the relations in the transport services market but is not subject to the market laws of supply and demand. Unlike transport costs, higher demand for transport services increases the external cost per transport unit; does not decrease them, respectively. All the activities in the transport sector that have negative implications on the environment and human health, while not being charged through the price of transport services, would have to be additionally charged. Finally, total transport costs increase and influence the competitiveness in the market of transport services. These changes are being accepted unwillingly. The very nature of these measures necessarily requires changes in consolidated, transport, logistics chains, and adaptation of business to new working principles, especially in the decision-making segment. The principle of using the shortest transport route from the starting point to the destination is not generally respected, and the length differences between the shortest and actual, utilized route are often several thousand kilometers. By promoting green logistics in transport, maritime and rail transport is promoted, and road transport is discouraged. This eliminates the external costs arising from congestion and accidents, both related to road transport. Implementation of the sustainable development principles in transport, as well as binding measures and activities, such as external costs in transport and their internalization, should influence the restructuring and optimization of freight flows, including the choice of traffic corridors.

## 6. Conclusions

According to the relative efficiency criterion, taking into account multiple variables, DEA enables the ranking of optimal traffic routes towards the same destination. Set in the function of principles of sustainability, DEA in this research model can determinate green corridors, the routes burdened with the least external costs among alternatives, but also optimal transport costs at the same time. A key

point of the model contribution is output settings, which consist of a unique and fixed variable of distance. The output is the given distance representing the shortest route between the starting point and destination, which was set following the principle of minimum energy consumption to achieve the same goal. If the current transport policy perseveres on the principles of sustainability, the given distance could become a driving concept of decision-makers in the transport logistic chain.

In the presented case study, the results show that green corridors from China to Central European destinations are traffic flows crossing the Northern Mediterranean and Adriatic ports, rather than the alternative Northern European seaports. Using DEA, an optimal balance among three input variables was determined for the given output respectively, a balance among transport cost, external cost, and transportation time optimal for the given distance on the chosen transport route marked as efficient. Traffic routes, assessed as inefficient, need corrections, usually a reduction in transport costs, external costs, and transport times, to achieve the efficiency of the selected route. Respecting the results obtained by the presented model enables the implementation of the sustainable transport policy by the minimal total cost and time of the transport. The higher the share of external costs in total transport costs, the more this and similar models will gain in importance as a tool for selecting the optimal traffic route. Results obtained under conditions of DEA settings specified for this research indicate that the direction of the green corridors could change traditional, consolidated freight flows, and promote formerly uncompetitive and ineffective ones. Qualified, trained, and motivated logistics operators as decision-makers in the logistics supply chain should get a central role and responsibility in this strategic shift.

Limitations are not related to the model, but the data set used. The model can accept any data set prepared in this way, for any amount of freight and any transport route. Due to its simplicity, the method is suitable for daily use. Future research should include more input variables on some other traffic routes following the proposed settings.

**Author Contributions:** L.V. developed the original idea for the study. L.V. was responsible for conceptualization, methodology, writing—reviewing, and editing of the article. T.P.J. conducted the formal analysis and was responsible for writing and preparation of the original draft version. G.G. took part in the article visualization and investigation while R.O. leads the article supervision and data validation. All authors have read and agreed to the published version of the manuscript.

**Funding:** This research received no external funding.

**Acknowledgments:** The authors wish to thank freight forwarders, logistic operators, maritime agents, industry experts, and other relevant stakeholders for participation in article creation and data provision.

**Conflicts of Interest:** The authors declare no conflict of interest.

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
