# Peer review of "Model of Determining the Optimal, Green Transport Route among Alternatives: Data Envelopment Analysis Settings"

_jmse, doi:10.3390/jmse8100735_

Round 1

Reviewer 1 Report

The article presents an interesting research topic related to the optimal, green transport route determination. The topic is up-to-date due to the growing importance of sustainability aspects and transport systems development. The writing style is understandable to reader.

However, there is a number of issues that should be improved:

  1. The motivation to undertake the research and the aim of the article should be added to article’s Abstract. The Abstract doesn’t describe clearly the content of the study.
  2. There is no specified Introduction Section with Authors motivation to undertake the research. The aim of the article wasn’t precisely defined. At one Section it is indicated that the “paper examines the competitiveness of intermodal, maritime-rail transport of TEU…”, another Section mentions that “study examines the relative efficiency of DMs on a traffic route…”. It seems that the developed methodology considers the case study (specific destination) analysis.
  3. There is no literature review. The in-depth literature analysis should be carried out. The Sections 1-3 of the article, that are placed before the “Materials and Methods” Section, cite only 7 literature positions. The relevant subject literature related to DEA implementation for solving transport tasks wasn’t sufficiently analyzed in the reviewed study. The article applies known method Data Envelopment Analysis (DEA) to solve the problem of determining the transport route. The novelty of the proposed approach wasn’t highlighted sufficiently.
  4. The selection of output and inputs to DEA may be also discussed. The Authors tried to justify the selection of these parameters. However, transport costs, time and distance are the well-known decision-making criteria for transport route selection. More and more research studies analyze the impact of external costs on decision-making related to transport of goods. Moreover, time of transport is influenced by distance and transport mean behavior under external (e.g. weather) conditions. It also has impact on transport costs, as it is well seen in maritime transport. Having only distance (defined also as “transport route”) as DEA output seems to be insufficient for decision-making.
  5. There are no Figures describing the case analyzed in the article (Efficiency frontier in DEA model with three-input and single-output variant).
  6. The Figure 1 doesn’t reflect precisely the described research idea (there is a lack of time and distance components).
  7. Each economic model has its mathematical basis. Therefore, they should be connected (opposite statement is in the text).
  8. Some contents in the article are almost repeated and there are  inaccuracies in the presentation of the content.

Author Response

Response to Reviewer Comments:

Point 1: The motivation to undertake the research and the aim of the article should be added to article’s Abstract. The Abstract doesn’t describe clearly the content of the study.

Response 1: We supplemented the abstract according to the suggestions

Point 2: There is no specified Introduction Section with Authors motivation to undertake the research. The aim of the article wasn’t precisely defined. At one Section it is indicated that the “paper examines the competitiveness of intermodal, maritime-rail transport of TEU…”, another Section mentions that “study examines the relative efficiency of DMs on a traffic route…”. It seems that the developed methodology considers the case study (specific destination) analysis.

Response 2: The introduction is formed and the goal of the paper is precisely emphasized

Point 3: There is no literature review. The in-depth literature analysis should be carried out. The Sections 1-3 of the article, that are placed before the “Materials and Methods” Section, cite only 7 literature positions. The relevant subject literature related to DEA implementation for solving transport tasks wasn’t sufficiently analyzed in the reviewed study. The article applies known method Data Envelopment Analysis (DEA) to solve the problem of determining the transport route. The novelty of the proposed approach wasn’t highlighted sufficiently.

Response 3: A review of the literature was written and the novelty of the proposed approach was highlighted

Point 4: The selection of output and inputs to DEA may be also discussed. The Authors tried to justify the selection of these parameters. However, transport costs, time and distance are the well-known decision-making criteria for transport route selection. More and more research studies analyze the impact of external costs on decision-making related to transport of goods. Moreover, time of transport is influenced by distance and transport mean behavior under external (e.g. weather) conditions. It also has impact on transport costs, as it is well seen in maritime transport. Having only distance (defined also as “transport route”) as DEA output seems to be insufficient for decision-making.

Response 4: The given distance in the output represents the essence of the work which is explained in detail. We agree that the input variables used are not the only ones that influence the choice of the optimal route, but they represent the main factors. The number of variables in the input is not limited and can be supplemented if necessary. That is also described in detail.

Point 5: There are no Figures describing the case analyzed in the article (Efficiency frontier in DEA model with three-input and single-output variant).

Response 5: A graphical representation of the DEA for the case study is not attached, but only for the DEA in general. The reason is that a graphical representation with 3 variables in the input requires a representation through 6 graphs, and so many graphs would obscure the essence of the work and become the essence itself.

Point 6: The Figure 1 doesn’t reflect precisely the described research idea (there is a lack of time and distance components).

Response 6: The flow diagram lists the EcoTransIT software that determines the distance and time values, and it is indicated in the text immediately before.

Point 7: Each economic model has its mathematical basis. Therefore, they should be connected (opposite statement is in the text).

Response 7: The mathematical expression is extended.

Point 8: Some contents in the article are almost repeated and there are inaccuracies in the presentation of the content.

Response 8: To provide the best possible explanation, there was sometimes a repetition. We tried to avoid that.

Reviewer 2 Report

The topic of the determining optimal traffic routes including external cost of transport presented in the manuscript ID: jmse-922097 entitled: Model of determining the optimal, green transport route among alternatives: Data Envelopment Analysis settings is important and very timely. The manuscript investigates this issue with appropriate methodological approach. The results of the study are significant and interpreted appropriately. However, the main weakness of the paper are introduction and the literature review. Here below, the authors can find some suggestions and observations to improve their manuscript:

  1. The paper is missing of a proper introduction. In the section 1 (Trends in European Union transport policy) the authors do not define the main purpose of the study and the current state of the research field has not been analysed. Generally the literature review is weak. Moreover, a high number of sentences in part 2 and 3 are not supported by citations (section 2 and 3, eg. line 76). My recommendation is to improve the literature review section (part 2 and 3) especially by adding what previous studies has found about the meaning of external costs in transport routes choice, and highlighting what emerge from your study.
  2. The conclusions should be more in depth. The main contribution to theory should be clear underlined. The managerial implications could be more in-depth as well. Moreover, no suggestions for future research are provided.
  3. Other minor comment: the authors used the Handbook on the external costs of transport, version 2014 (line 109). I suggest to update the source to the new version from January 2019.

Author Response

Response to Reviewer Comments

Point 1: The paper is missing of a proper introduction. In the section 1 (Trends in European Union transport policy) the authors do not define the main purpose of the study and the current state of the research field has not been analysed. Generally the literature review is weak. Moreover, a high number of sentences in part 2 and 3 are not supported by citations (section 2 and 3, eg. line 76). My recommendation is to improve the literature review section (part 2 and 3) especially by adding what previous studies has found about the meaning of external costs in transport routes choice, and highlighting what emerge from your study.

Response 1: An Introduction with a literature review was written

Point 2: The conclusions should be more in depth. The main contribution to theory should be clear underlined. The managerial implications could be more in-depth as well. Moreover, no suggestions for future research are provided.

Response 2: The conclusion is expanded and supplemented with elements that you have recommended.

Point 3: Other minor comment: the authors used the Handbook on the external costs of transport, version 2014 (line 109). I suggest to update the source to the new version from January 2019.

Response 3: The new edition of the source of unit prices of external costs in transport could not be used because it was published too late, and all other costs refer to the year 2018. This is now pointed out in the text.

Reviewer 3 Report

This is overall an interesting paper but I feel it is not placed accurately within the existing literature. There are many papers that have used EconTransit to compare alternatives (see for example Panagakos et al. from 2014 - there are many papers published in the last 10 years actually) and many more have incorporated the external costs from the EU handbook .
The authors should include a review of some of these papers.

2. Then the use of EcoTransit itself should be discussed focusing on its limitations. Using DEA to compare alternatives is indeed an interesting work but the results of any analysis is as good as its inputs. Do the authors feel that, say the total times provided are real or realistic ? Do they feel that the EcoTransit model has all container schedules and connections integrated ?
Off course not. The manual (see link below) says for instance "EcoTransIT World calculates direct port to port relations on actual sea routes. As most of these relations do not exist in reality...". So the times are actually the quickest possible if there is direct connection In addition, say the container arrives in Thessaloniki; how is it going to be transported to Budapest? Will there be a truck waiting for it, or a cargo train will be used (in the latter the cargo might have to wait for some time or even days). These are well known limitations to the EcoTransIT model that need to be discussed. EcoTransIT is very popular because it is easy to use but this is the reason why it also exhibits all these flaws.

3. Now, regarding the use of social costs - these need to be updated with the figures provided in the 2019 EU handbook - the authors use here the previous versions.
They need to check if the relevant figures have been updated and use the recent ones; see discussion bellow.

4.  Sections 1,2, and 3 are too short. I suggest merging some - but there is definitely a need to expand the literature review. Not sure I agree with the statement That "external costs have not been sufficiently considered" (line 65)
The authors actually refer to the EU handbook on external costs (see their Ref. #6 and 7. There is actually an updated Handbook (the 2019 version see https://ec.europa.eu/transport/sites/transport/files/studies/internalisation-handbook-isbn-978-92-79-96917-1.pdf ). I suggest the authors add this updated version and review a number of relevant papers (also cited in the Handbook). 

2. The methodology is simplistic but I feel acceptable. As per my comment above the 2019 Handbook values should be used to produce more updated results. There is also a need to discuss the practical applicability of the model. Are we considering the transportation of a single container? Because if we want to transport 100 containers then the solution through Thessaloniki or any other point will require the use of a huge number of trucks or if trains are used then the capacity constrains should be looked at.

REFERENCES

Panagakos, G. P., Stamatopoulou, E. V., & Psaraftis, H. N. (2014). The possible designation of the Mediterranean Sea as a SECA: A case study. Transportation Research Part D: Transport and Environment, 28, 74–90. doi:10.1016/j.trd.2013.12.010 

2019 Handbook on the external costs of transport
https://ec.europa.eu/transport/sites/transport/files/studies/internalisation-handbook-isbn-978-92-79-96917-1.pdf

EcoTransIT manual
https://www.ecotransit.org/download/EcoTransIT_World_Methodology_ShortVersion_2019.pdf

Author Response

Response to Reviewer 1 Comments

Point 1: This is overall an interesting paper but I feel it is not placed accurately within the existing literature. There are many papers that have used EconTransit to compare alternatives (see for example Panagakos et al. from 2014 - there are many papers published in the last 10 years actually) and many more have incorporated the external costs from the EU handbook .
The authors should include a review of some of these papers.

Response 1: A review of the literature has been written

Point 2: Then the use of EcoTransit itself should be discussed focusing on its limitations. Using DEA to compare alternatives is indeed an interesting work but the results of any analysis is as good as its inputs. Do the authors feel that, say the total times provided are real or realistic ? Do they feel that the EcoTransit model has all container schedules and connections integrated?
Off course not. The manual (see link below) says for instance "EcoTransIT World calculates direct port to port relations on actual sea routes. As most of these relations do not exist in reality...". So the times are actually the quickest possible if there is direct connection In addition, say the container arrives in Thessaloniki; how is it going to be transported to Budapest? Will there be a truck waiting for it, or a cargo train will be used (in the latter the cargo might have to wait for some time or even days). These are well known limitations to the EcoTransIT model that need to be discussed. EcoTransIT is very popular because it is easy to use but this is the reason why it also exhibits all these flaws.

Response 2: We couldn't discuss EcoTransIT software because that's not the topic of the paper. This program is only an aid that can but does not have to be used in the model. The authors are very aware of the limitations of this program, but they also tested it over a year and were surprised at how sophisticated and detailed it is, and at the same time easy to use. e.g. the program takes into account railway sections that are not electrified! It also takes into account truck transport if direct transshipment from ship to train is not possible. However, this software is only an aid, and its limitations are related to all routes. We have now included these explanations in the text.

Point 3: Now, regarding the use of social costs - these need to be updated with the figures provided in the 2019 EU handbook - the authors use here the previous versions.
They need to check if the relevant figures have been updated and use the recent ones; see discussion bellow.

Response 3: The social cost update could not be used because it was published too late to be relevant to face other costs calculated for 2018

Point 4: Sections 1,2, and 3 are too short. I suggest merging some - but there is definitely a need to expand the literature review. Not sure I agree with the statement That "external costs have not been sufficiently considered" (line 65)
The authors actually refer to the EU handbook on external costs (see their Ref. #6 and 7. There is actually an updated Handbook (the 2019 version see https://ec.europa.eu/transport/sites/transport/files/studies/internalisation-handbook-isbn-978-92-79-96917-1.pdf ). I suggest the authors add this updated version and review a number of relevant papers (also cited in the Handbook). 

Response 4: A new Introduction has been formed and line 65 has been corrected.

Point 5: The methodology is simplistic but I feel acceptable. As per my comment above the 2019 Handbook values should be used to produce more updated results. There is also a need to discuss the practical applicability of the model. Are we considering the transportation of a single container? Because if we want to transport 100 containers then the solution through Thessaloniki or any other point will require the use of a huge number of trucks or if trains are used then the capacity constrains should be looked at.

Response 5: EcoTransIT has been tested concerning the amount of cargo, and the final results of the DEA with the external costs of the higher amount of cargo included were the same. We made sure that EcoTransIT takes care of the capacity of the ship, train, and truck. Therefore, for the sake of clarity, we decided to make calculations per unit freight. Discounts on quantity and insurance are not taken into account, but they are approximately the same for all routes. We have included the explanation in the text. Recommended literature is also included.

Round 2

Reviewer 1 Report

The reviewed article was significantly improved and the doubts were cleared up.

Reviewer 3 Report

I would like to thank the authors for taking the time to address my earlier comments. Many of them were addressed (e.g the literature review part has been improved) in their revised manuscript but I feel the important points have not been addressed.

The paper is an interesting area but there are already some papers in the literature that use DEA  or other methods (e.g. multicriteria decision analysis tools) to evaluate alternative routes.
Although the methodology is not original it could be an interesting case study. 

In that case thought the inputs (economic cost, social costs and time ) should be clearly stated. In table 1 we could see some inputs but their calculation is not clear.
1. Time (and distance) are calculated using EcoTransit. I understand there are other ways to calculate them but the authors are using this specific software therefore (as I asked in my original review the way they were claculated and the limitations should be discussed. EcoTransit has 2 ways to do calculations, in the detailed one for example the average weight of containers is used. You could also put in an appendix the modes selected for each route, ie. which part was sea which road so that we could have a better idea of your case study.
As I mentioned before transit time the existence of routes is not checked by the calculator. Now if the service does not exist or there is need for transipment (as I explained in the case of Thessaloniki for which I am very familiar with) then the days that you state in Table 1 are not correct. I understand that is what the software is providing you with but if your inputs are wrong then the results are wrong. Therefore the uncertainties should be discussed.

2. I am coming now to the external costs. You need to discuss more what exactly is included in the them. Econtransit can calculate CO2 , SOx, PM emissions and so on. Which ones did you include. What was the per unit values that you have used as the the 2014 Handbook. If these values are not different compared to the 2019 Handbook then you do not need to run your DEA software again. Or if they are close. But if they are not then the route ranking will change. Note that you claim you present work done in 2018, the 2019 handbook was published in January 2019. If the values are substantially different then I might claim that your results are obsolete.

(Not to mention that the same could be said for the freight rates ). At the same time we know that rates and other inputs are volatile. The reader should be able to see all the inputs so to have a rough idea how relevant the results might be.

I understand that the authors avoid all the above by focusing more on the model that can accept any input. I agree if that is the case -the case study then is not much important as dated data are used (2018 data and social costs based on the 2014 handbook). If this is the case, what is the advantage of using DEA instead of a MCDA method ?

Besides, what is the realistic use of this model: The shipper or the logistics provider as the author claim? The shippers, in general, do not take the social costs into their decision making unless say they pay a tax on the CO2 they produce. Logistics providers (say a big forwarder) could consider the societal costs in order to be greener - although literature suggests that it is the total cost that is important for them.

The authors should consider these points, assuming they want to focus more on the method then I suggest the compare their model with other approaches and consider its practical applicability better.

There are many papers benchmarking freight routes with data envelopment analysis (DEA) some of them coupled with other techniques
see for example below
https://ageconsearch.umn.edu/record/291196/files/Abstracts_19_05_15_17_38_16_35__143_107_210_16_0.pdf

Round 3

Reviewer 3 Report

I thank the authors for their effort to address my comments during the past rounds of revision. I feel that most of my concerns have been addressed.